# The Diagnostic Performance of a Clinical Diagnosis of Diabetic Kidney Disease

**DOI:** 10.3390/life13071492

**Published:** 2023-06-30

**Authors:** Ken-Soon Tan, Stephen McDonald, Wendy Hoy

**Affiliations:** 1School of Medicine (Centre for Chronic Disease), University of Queensland, Brisbane 4072, Australia; 2School of Medicine and Dentistry, Griffith University, Southport 4222, Australia; 3Adelaide Medical School, University of Adelaide, Adelaide 5000, Australia; 4ANZDATA Registry, Adelaide 5001, Australia

**Keywords:** diabetes mellitus, chronic kidney disease, biopsy, proteinuria

## Abstract

Background: Diabetic kidney disease (DKD), a common cause of CKD and kidney failure, is usually diagnosed clinically. However, there is little evidence comparing the performance of a clinical diagnosis to biopsy-proven diagnosis. Purpose of the study: Diagnostic performance of a clinical diagnosis was determined in a group of patients with diabetes and chronic kidney disease who underwent kidney biopsy after an initial clinical diagnosis. Methods: A data analysis of 54 patients who were part of a study cohort for a prospective analysis of cardiovascular and kidney outcomes and who had undergone kidney biopsy after an initial clinical diagnosis of DKD or non-DKD (NDKD) at enrolment was used. We determined the sensitivity, specificity, and positive and negative predictive values of a clinical diagnosis of DKD. Results: A total of 37 of 43 patients clinically diagnosed with DKD also had biopsy-proven DKD, whilst only 1 of 11 patients who had clinically diagnosed NDKD had biopsy-proven DKD. Sensitivity was 97.4%, specificity was 62.5%, positive predictive value 86%, and negative predictive value 90.9%. Comparable values were obtained when analysis was restricted to those with primary rather than secondary diagnosis of DKD or when restricted to those with only DKD found at biopsy. Conclusion: A clinical diagnosis of DKD has high sensitivity and is unlikely to overlook cases but may lead to overdiagnosis.

## 1. Introduction

Diabetic kidney disease (DKD) remains a common complication of diabetes mellitus (DM) and is a common cause of both chronic kidney disease (CKD) and kidney failure worldwide. Given the increasing prevalence of DM (especially DM type 2) worldwide [1], the prevalence of CKD in association with DM is expected to increase in tandem. The diagnosis of DKD is usually made based on clinical rather than histopathological findings. Indeed, current American diabetic association (ADA) guidelines [2] state that the diagnosis of diabetic kidney disease (DKD) “should usually be made clinically based on the presence of albuminuria and/or reduced eGFR in the absence of features suggesting other primary causes of kidney damage”. A longer duration of DM and the presence of diabetic retinopathy support a diagnosis but are not required. The ADA guidelines [2] suggest considering alternative kidney diagnoses if there is rapidly increasing albuminuria/nephrotic syndrome or rapidly decreasing GFR. They also acknowledge the increasing prevalence of non-albuminuric DKD. Thus, the clinical diagnosis of DKD is not as straightforward as it might first appear. DKD can co-exist with other primary kidney diseases.

Nonetheless, kidney biopsy still has a role in the clinical management of DKD and is certainly still performed in patients with DM and kidney disease. It remains the “gold standard” for the diagnosis of DKD and helps to establish prognosis by determining the severity of disease. Renal pathology society has developed a histopathological classification consensus of diabetic kidney disease applicable to those with DM types 1 and 2 [3]. 

Given the reliance on a “clinical diagnosis” of DKD, there is little published evidence on its diagnostic performance (sensitivity, specificity, positive predictive value, negative predictive value, and accuracy) compared to the “gold standard” histopathological diagnosis.

In a pooled meta-analysis of 48 studies involving 4876 diabetic patients (where histopathological diagnosis was made), Fiorentino et al. [4] determined “positive predictive values” for the diagnosis of DKD alone, DKD co-existing with another pathology (mixed disease), or another primary pathology (NDKD). The positive predictive values were calculated from a random effects model presumably based on associations with clinical characteristics noted after biopsy diagnosis was obtained. Values were low for all 3 types of pathologies (at best, 69% for the presence of DKD, whether as sole or shared pathology). The authors also noted that the presence of NDKD (whether as sole or shared pathology) in diabetic patients was more common than previously believed. However, on reviewing the methods sections of each included study, clinical diagnoses were only officially assigned prior to histopathological diagnosis in only one study [5]. In this study, the gold standard histopathological diagnosis was also made post-mortem in the majority. In one other study [6], kidney biopsies were performed in patients with “presumed diabetic nephropathy”, although in these patients, the diagnosis of “presumed diabetic nephropathy” had been conferred by their primary care physician rather than a nephrologist, and biopsy had subsequently been performed by the nephrologist because there was sufficient suspicion of an alternative diagnosis.

More recently, Tong et al. [7] performed a review of the global renal biopsy literature in patients with type 2 DM over the period of 1977 to 2019 inclusive. Their aim was to identify and review studies which reported on clinical indicators to distinguish DKD from NDKD. A total of 40 studies involving 5304 patients met the criteria. They found that the most common clinical indicators were diabetic retinopathy and the duration of diabetes.

We performed a separate literature search across two databases (MEDLINE and EMBASE), dating as far back as 1976, looking for studies where the diagnostic performance of a pre-assigned clinical diagnosis of DKD was compared against a subsequent histopathological diagnosis. This search yielded only the same study mentioned above [5].

It is useful to know whether DKD is present (even if co-existent with another pathology) because this will certainly help guide prognosis. If DKD is present, then even specific therapy for co-existent NDKD will not fully reverse kidney damage as the DKD will still be present. In some cases, specific treatment might even accelerate the progression of DKD, e.g., steroid or calcineurin inhibitor therapy for certain glomerulopathies may worsen DM and BP control, both of which will accelerate the progression of the co-existing DKD.

Even when DKD is believed the sole pathology, there is still merit in being certain that this is indeed the case, because once again, the prognosis might also differ significantly from the situation where NDKD is present, e.g., SGLT-2 inhibitor therapy might be less effective if NDKD co-exists. In this study, our purpose was to determine the diagnostic performance of a clinical diagnosis of DKD in a series of patients who underwent kidney biopsy after first being assigned a clinical diagnosis.

## 2. Materials and Methods

### 2.1. Patient Population

The patients were all enrolled in the CKD.QLD registry. This is the first formal Australian registry of non-dialysis CKD patients receiving public hospital kidney specialist care [8,9] and has enrolled patients receiving public hospital outpatient kidney specialist care at one of eleven participating hospitals in the state of Queensland, Australia, through a process of written informed consent since its inception in 2011 (ethics approvals HREC/15/QRBW/294 and University of Queensland 2011000029). The registry collects data on routine clinical care with no influence on the nature or delivery of that care (i.e., a pragmatic record of the clinical course). To determine the diagnostic performance of a clinical diagnosis of DKD, we retrospectively analysed data from the subset of patients with DM enrolled between registry inception and 31 December 2016 who had undergone kidney biopsy only after the initial clinician assigned diagnosis at enrolment. The censor date for this analysis was 31 December 2017, so that all patients had at least 12 months after enrolment to undergo kidney biopsy.

### 2.2. Diagnosis of DKD and NDKD

#### 2.2.1. Clinical Diagnosis of DKD and NDKD

At the time of enrolment, clinicians assigned at least one diagnosis for the underlying CKD, based on the primary renal disease coding system utilized by the Australian and New Zealand dialysis and transplant registry (ANZDATA) [10]. On this basis, nearly all patients (>99%) were assigned at least one clinical diagnosis for their CKD at time of enrolment.

If more than one clinical diagnosis was assigned, we considered the order as hierarchical, and clinically diagnosed DKD was deemed present if DKD was among the first two listed diagnoses (if more than one was assigned). All other patients were considered to have clinically diagnosed NDKD.

#### 2.2.2. Biopsy Diagnosis of DKD and NDKD

All kidney biopsies performed within Queensland Health require written informed consent from the patient. Biopsies are performed by nephrologists (or specialty trainees) under ultrasound guidance. Tissue cores are sent for analysis via light microscopy, immunofluorescence, and electron microscopy. The reporting of kidney biopsies for the pre-specified study period (2011–2017) was performed by renal pathologists at the state capital’s two tertiary hospitals.

DKD was defined as present if there were histopathological changes as per the renal pathology society consensus guidelines [3], even if the DKD was a co-existing pathology. A biopsy diagnosis of “NDKD only” was made if there were no histopathological features of DKD in the biopsy specimens.

### 2.3. Data Analysis

We calculated the sensitivity, specificity, positive predictive value, negative predictive value, and accuracy of a clinical diagnosis of DKD against the “gold standard” of a biopsy diagnosis.

### 2.4. Statistical Analysis

Summary statistics were number (%), mean (+/− SD), or median (IQR). Comparisons of proportions were carried out using the chi^2^ test or Fisher’s exact test. A 95% CI for sensitivity and specificity were calculated using the binomial (“Clopper–Pearson”) method. A 95% CI for positive predictive value, negative predictive value, and diagnostic accuracy was calculated using the method of Mercaldo et al. [11]. The latter depends on a prevalence estimate for DKD (whether as sole or co-existing pathology). Using weighted data from the meta-analysis of Fiorentino et al. [4], we estimated this prevalence to be 63%.

## 3. Results

### 3.1. Overview of Study Population and Flow

Among the 2370 patients with DM enrolled in the registry during the specified period, 185 (7.8%) had undergone kidney biopsy although of these, only 54 had undergone kidney biopsy after their initial clinical diagnosis/diagnoses. All 54 biopsies yielded sufficient tissue for a histopathological diagnosis to be assigned. Figure 1 summarizes the STARD diagram of the study flow.

### 3.2. Baseline Characteristics

The baseline characteristics of these 54 patients and listed indications for biopsy are presented in Appendix A. In summary, the mean age was 58.2 years (SD 11.5), 50% of patients were male, 59.3% were on Insulin (+/− oral hypoglycaemic agent [OHA]), 33.3% were on OHA only, the median CKD-EPI eGFR was 38.5 mL/min/1.73 m^2^ (IQR 29, 50), 85% of patients had proteinuria (urine ACR > 30 mg/mmol or protein:creatinine >50 mg/mmol), and >90% were already on RAAS blocker therapy (Table 1). It is of note that despite the high RAAS blocker therapy rate, nearly half the patients (48.1%) still had nephrotic range proteinuria (Table 1). In 43 patients (79.6%), DKD was among the first two allocated clinical diagnoses prior to biopsy. The most common indication for biopsy was to exclude a co-existing glomerulonephritis (GN), largely driven by the presence of microscopic haematuria.


**Cardiac:**
Acute coronary syndrome.Coronary revascularization event—elective or emergency (but not including diagnostic angiogram).Heart failure.

**Cerebrovascular:**
Cerebrovascular accident (not including transient ischaemic attack).Carotid artery revascularization (elective or emergency).

**Other vascular:**
Gangrene requiring limb/digit amputation.Peripheral artery revascularization procedure (elective/emergency) not including diagnostic angiogram.Aortic aneurysm hospitalization for rupture or leak or repair (elective or emergency)—not including incidental discovery during admission or outpatient clinic review.Bowel infarction—confirmed via the histology of resected bowel or relevant history and imaging (if managed conservatively).


### 3.3. Biopsy Diagnoses

All 54 biopsy specimens yielded sufficient tissue for histopathological diagnosis. A total of 38 patients (70.3%) had evidence of DKD at biopsy. A total of 14 of these (25.9%) had co-existing pathologies present (i.e., DKD + NDKD).

Co-existing pathologies in the 14 patients are listed in Table 2. The most common co-existing pathology was secondary focal segmental glomerulosclerosis (FSGS).

A total of 16 patients only had evidence ofNDKD on biopsy. Their biopsy diagnoses are listed in Table 3.

### 3.4. Diagnostic Performance of Clinically Diagnosed DKD

The clinical diagnosis and subsequent biopsy findings are compared in Table 4.

Based on the information in Table 4, sensitivity was 97.4% (95% CI 86.2–99.9%) and specificity was 62.5% (95% CI 35.4–85.8%). Based on the estimated prevalence of DKD (see statistical methods section), the positive predictive value was 81.6% (95% CI 66.8–91.7%), the negative predictive value was 93.3% (95% CI 61.8–99.9%), and diagnostic accuracy was 84.7% (72.0–92.0%).

We performed two sensitivity analyses:Excluding four patients with a second listed clinical diagnosis of DKD (presuming that these were patients in whom there was less confidence in the diagnosis)—the results are summarized in Table 5.Sensitivity (97.1%), positive predictive value (83.2%), negative predictive value (93.2%), and diagnostic accuracy (85.9%) were essentially unchanged, although specificity (66.7%) had improved somewhat.Excluding the 14 patients who had DKD as a shared pathology at biopsy—the results are summarized in Table 5.Sensitivity and negative predictive value had increased to 100%, whilst positive predictive value had improved to 86.1%. Specificity remained steady at 62.5%, as did diagnostic accuracy at 86.1%.

### 3.5. Associations of Biopsy Diagnosis with Proteinuria

We examined the association between biopsy diagnosis (DKD or only NDKD) and proteinuria level at enrolment. These results are summarized in Table 6. A greater proportion of patients with DKD had ACR > 30 mg/mmol (Categories 3 and 4) than those with NDKD only (*p* < 0.05). Furthermore, nephrotic range proteinuria (Category 4) was more common in those with DKD (*p* = 0.001). The latter remained true even when analysis was restricted to those with only DKD or NDKD at biopsy (*p* < 0.005).

### 3.6. Associations of Biopsy Diagnosis with Microscopic Haematuria

We examined the association between biopsy diagnosis (DKD or NDKD only) and microscopic haematuria (as mentioned in the indication for biopsy, Appendix A). These results are summarized in Table 7. In our cohort, a greater proportion of patients with DKD present had microscopic haematuria (36.8%) than those with pure NDKD (18.8%), although the difference was not statistically significant (*p* = 0.19).

## 4. Discussion

This study was conceived when the analysis of pragmatic data collected during routine clinical care for our principal study revealed that some patients with clinician-labelled diagnoses for their CKD had subsequently undergone kidney biopsy, thus presenting an opportunity to assess the diagnostic performance of a clinical diagnosis of DKD. We had no influence on the decision or indications to perform kidney biopsy in these patients.

Our results show that in this care setting, the sensitivity of clinical diagnosis for the presence of DKD was very high (i.e., unlikely to miss the diagnosis of the condition) and it did not seem to matter whether the clinical diagnosis of DKD was first or second listed (i.e., no obvious effect of diagnostic hierarchy). The specificity was lower, suggesting a tendency to “over-diagnose” the condition. However, given the higher prevalence of DKD compared to NDKD in this population, the negative predictive value was also high, i.e., “no likely means no”. The overall accuracy of clinical diagnosis was >80%. The diagnostic performance was similar even when the question being asked was “is DKD the only pathology present?” rather than “is DKD present?”.

Multiple studies in patients with type 2 DM have examined the association of various clinical parameters with the presence of DKD or NDKD on histopathology. In the systematic review by Tong et al. [7], the five most common predictors of DKD or NDKD in order of frequency, were: diabetic retinopathy, the duration of diabetes, haematuria, proteinuria, and HbA1c. The other principal microvascular complication of neuropathy was not listed.

In six studies published since the that systematic review [15,16,17,18,19,20], the most common shared predictors were the presence/absence of diabetic retinopathy [15,16,17,19,20], the duration of diabetes [16,17,18], and the presence of haematuria [18,20]. A shorter duration of DM (<5 years) predicted NDKD [17,18], whilst longer duration predicted DKD [16]. This latter association may underlie the common assumption that proteinuria in patients with DM1 represents DKD. The presence of diabetic retinopathy predicted the presence of DKD [16,19], whilst its absence predicted NDKD [15,17,20]. The presence of haematuria predicted NDKD [18,20]. Lower HbA1c was a significant predictor of NDKD in one study [18] but was not a significant predictor in another [19]. Peripheral neuropathy was not a significant predictor of DKD in one study [16].

However, these studies examined the performance of individual factors rather than that of an overall clinical impression, which considers the impact of multiple factors. Thus, the authors cannot claim to examine the diagnostic performance of a clinical diagnosis. The associations were determined in retrospective fashion (i.e., determined after biopsy diagnoses were available) and clinically significant associations were not then explored in a prospective fashion (i.e., a pre-test probability of DKD attributed to the presence of the clinical factor was not then confirmed via a subsequent biopsy diagnosis). Furthermore, the associations were weak in two studies [15,16], limiting their clinical utility. Finally, all but one study [18] involved patients from single centres. In the only multi-centre study, Chemouny et al. [18] explored the associations of clinical features with a subsequent biopsy diagnosis of NDKD in 463 diabetic patients from four hospitals in France who specifically underwent kidney biopsy to exclude NDKD. Only 187 patients (40%) had NDKD (either as a sole pathology or in combination with DKD). In their multivariate regression model, the authors identified a diabetes duration of <5 years, a urine protein:creatinine ratio of <300 mg/mmol, CKD-EPI eGFR of > 15 mL/min, and haematuria as significant predictors of NDKD. Curiously, “unknown” diabetic retinopathy status was also a significant predictor, although neither non-proliferative nor proliferative retinopathy status were negative predictors of NDKD. Given that all patients had undergone biopsy because NDKD was suspected, the prevalence of NDKD at biopsy would suggest that the diagnostic performance of a clinical diagnosis of NDKD is poor.

Our multi-centre study enrolled patients from 11 different hospitals across the state, which should address the issue of site-specific clinical practice. Whilst the data collected by the CKD.QLD registry did not allow us to assess the impact of diabetic retinopathy or duration of diabetes, it is likely that enrolling clinicians would have considered the presence of these factors when assigning a clinical diagnosis of DKD, given the widespread and long-held acceptance of both as predictors of DKD.

Three recent studies [21,22,23] have examined the associations of various clinical features with a kidney biopsy diagnosis and have used these associations to develop predictive models. Diabetic retinopathy was the only clinical feature which was included in all predictive models. The duration of diabetes was explored in one model [21] but was not significant. Once again, these associations were retrospectively determined after obtaining a biopsy diagnosis, and predictive models were not prospectively tested. These studies all involved single centres, Asian populations, and focused only on patients with type 2 DM. It is notable that two of these studies specifically involved patients with proteinuria [21,23], who would normally be assumed to have DKD, according to ADA criteria [2]. 

To our knowledge, only one prior study [5] has genuinely compared the diagnostic performance of the clinical diagnosis of DKD against the gold standard of histopathological diagnosis in the same patient. Biesenbach et al. [5] examined the diagnostic performance of a clinical diagnosis of DKD and vascular disease compared to a histopathological diagnosis in 84 patients with DM2, who subsequently progressed to kidney failure and dialysis, i.e., all patients had been allocated a clinical diagnosis of either DKD or vascular disease prior to commencing dialysis. In their study, 14 patients had undergone kidney biopsy whilst alive, but the remaining 70 had their histopathological diagnosis confirmed post-mortem. The authors did not clarify whether the 14 patients who had undergone kidney biopsy had done so before or after clinical diagnosis was assigned. The authors did not differentiate between diagnoses obtained pre- or post-mortem and the results were reported for the whole group. For a clinical diagnosis of DKD, sensitivity was 95%, specificity was 78%, positive predictive value was 84%, and negative predictive value was 82%. Except for specificity, our results are similar. However, Biesenbach et al. [5] limited their subjects to patients with a poor CKD prognosis (i.e., all patients developed kidney failure and required dialysis), so there may have been more selection bias than our study. Nonetheless, the authors could genuinely claim that their decision to obtain a histopathological diagnosis had no bearing on patient outcomes.

Although the ADA guidelines [2] have suggested that the presence of nephrotic range proteinuria should raise suspicion of NDKD, our results indicate that this does not necessarily rule out the co-existence of DKD. Indeed, we found that nephrotic proteinuria (Category 4) was far more common in those with DKD present at biopsy compared to those with only NDKD present.

We also found that microscopic haematuria was more commonly reported in patients with DKD at biopsy, although the difference did not achieve statistical significance.

In the DAPA-CKD study, CKD patients without DM (and therefore no DKD) derived similar kidney outcome benefits from SGLT2 inhibitor therapy to patients with DM2 [24]. These findings might suggest that there is no point diagnosing the cause of CKD, as there is now an effective treatment which benefits all causes of CKD. This insinuation is worth scrutiny. In DAPA-CKD, patients without DM (i.e., no DKD) comprised ~1/3 of the study population, and 13.6% of patients with DM2 were not allocated a clinical diagnosis of DKD [25]. Overall, ~20% of patients (mainly those without DM) had undergone biopsy [25], i.e., the majority (especially those with “DKD”) were clinically diagnosed. Whilst a pre-specified subgroup analysis [26] has reported that kidney outcome benefits were similar, regardless of whether patients had DKD, GN, hypertensive/ischaemic nephrosclerosis, or a kidney disease of uncertain aetiology, the hazard ratios for the latter two groups had 95% confidence intervals which included 1 (i.e., not statistically significant). Furthermore, the investigators in this trial did not consider the impact of multi-pathology (i.e., more than one kidney diagnosis). Thus, we cannot be certain that the benefits of these agents are quite as universal as touted, and there is still merit in making accurate diagnoses in CKD, especially in DM, where assumptions are more likely to be made.

### Limitations

This is a small patient sample. Furthermore, the small proportion of patients with type 1 DM mean that we cannot comment on the diagnostic performance of a clinical diagnosis in these patients. However, the sample size in not much smaller than some recently published studies.

Whilst indication bias is likely to be present, this is no different from most other published studies on kidney biopsy in patients with DM, where the majority had been performed because of suspected NDKD. Our principal intent was determining the accuracy of a prior allocated clinical diagnosis of DKD rather than to determine indications for the biopsy. In any case, ~80% of our patients had already been clinically diagnosed with DKD (in 55% this was their primary clinical diagnosis) prior to biopsy.

Biopsy safety was not explored here as it was not part of the study intent. However, no major bleeds were recorded from these 54 biopsies. Although the lack of bleeding complications could be attributed to selection of “lower risk” patients for biopsy, the rate of significant bleeding complications is generally quoted as ~1% [27], such that our sample size might have been too small (i.e., a type 2 error). Furthermore, at least a quarter of patients were at CKD stage 4+ and would have been deemed at higher risk of bleeding post-biopsy.

## 5. Conclusions

We have shown that clinician-labelled diagnosis of DKD in the kidney specialist care setting has high sensitivity and seems unlikely to miss diagnosing cases. However, there is a tendency towards overdiagnosis (lower specificity). Caution should also be exercised, as non-proteinuric DKD is likely to be more prevalent in the kidney specialist care setting. Thus, we propose a larger pragmatic study to further investigate this. Adult patients with DM, who have any of the standard indications for native kidney biopsy in adults [28] (i.e., nephrotic syndrome, a combination of unexplained haematuria and proteinuria, AKI that is not pre-renal or obstructive, or early unexplained CKD) and who provided informed consent for the procedure and reporting of their data are first allocated a clinical diagnosis of either DKD, mixed disease, or NDKD. The diagnostic performance of the clinical diagnosis is then reviewed after biopsy.

## Figures and Tables

**Figure 1 life-13-01492-f001:**
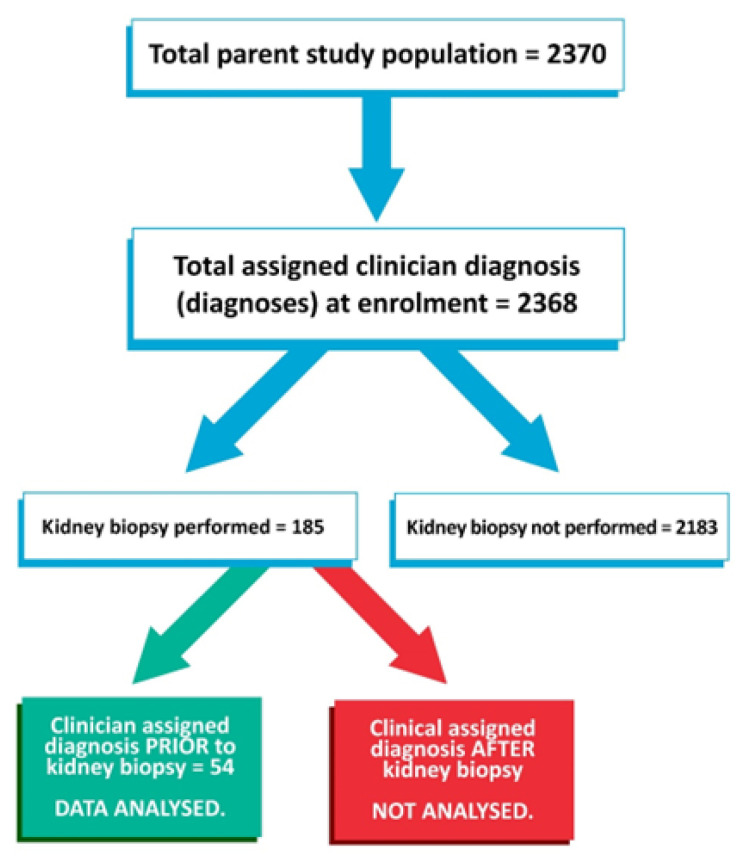
STARD diagram for the study flow.

**Table 1 life-13-01492-t001:** Baseline characteristics of 54 patients who underwent kidney biopsy after initial clinical diagnosis.

Parameter	Summary Statistic Value
**Age (years)**	Mean 58.2 (SD 11.5)
**Gender**	Male = 27 (50%)
**BMI** (information available for 46 patients)	Median 32.1 (IQR 30.2; 36.7)
**Treatment for DM**	
Insulin (+/− ^a^ oral hypoglycaemic agent (OHA))	32 (59.3%)
OHA only	18 (33.3%)
Diet only	4 (7.4%)
**DM type 1**	6 (11.1%)
**HbA1c**	Median 7.4% (IQR 6.6; 8.8)
**CKD-EPI eGFR (mL/min/1.73 m^2^)**	Median 38.5 (IQR 29; 50)
**^b^ Proteinuria category**	
1 (albumin:creatinine (ACR) < 3 or protein:creatinine < 15 mg/mmol)	4 (7.4%)
2 (ACR ≥ 3 but <30 or protein:creatinine ≥ 15 but <50 mg/mmol)	4 (7.4%)
3 (ACR ≥ 30 but <220 or protein:creatinine ≥ 50 but <300 mg/mmol)	20 (37.0%)
4 (ACR ≥ 220 or protein:creatinine ≥ 300 mg/mmol)	26 (48.1%)
**^c^ RAAS blocker therapy**	50 (92.6%)
**Clinical diagnosis includes DKD**	43 (79.6%)
**Hypertension** (information available for 52 patients)	44 (84.6%)
**^d^ Previous history of MACE at enrolment**	15 (27.8%)

^a^ SGLT-2 inhibitor therapy was first approved for pharmaceutical benefits subsidy (PBS) in Australia as third line oral hypoglycaemic agent therapy after sulphonylureas and metformin from 2013 [12]. However, in this initial approval, access was restricted to those with HbA1c > 7%, concurrent therapy with either DPP or insulin was precluded, as was triple therapy with metformin and sulphonylurea. Initially, caution was advised in those with eGFR < 60 mL/min/1.73 m^2^ [13]. All these restrictions and cautions have since been removed and amendments to PBS criteria following evidence of cardiovascular and kidney outcome benefit have been added, although these latter additions only came into effect in September 2022 (i.e., well after the study period). Thus, although no information was available in the registry regarding the use of these agents in the 54 patients, it is unlikely to have been high. Of note, the only GLP-1 agonist available in Australia during the study period was Exenatide, and its use was limited to the clinical trial setting. ^b^ Proteinuria category was determined from either spot urine albumin:creatinine or protein:creatinine ratios. As there was no consistency in the measurement methods, a category system was used, as detailed above. Categories are based on the KDIGO 2012 clinical practice guideline for the evaluation and management of chronic kidney disease [14] with sub-division of the A3 category into a non-nephrotic and nephrotic range (category 4). The threshold for nephrotic range proteinuria was also taken from the KDIGO guideline (specifically, page 29 of that document) [14]. ^c^ RAAS blocker = Renin–Angiotensin–Aldosterone system blocker, which includes mineralocorticoid antagonists. ^d^ MACE was classified under three main headings.

**Table 2 life-13-01492-t002:** Co-existing pathologies in 14 patients who had DKD and NDKD at biopsy.

Co-Existing Pathology	*n* (Total = 14)
Secondary FSGS	5
IgA nephropathy	2
ANCA associated vasculitis (microscopic polyangiitis).	2
Membranoproliferative/mesangiocapillary GN (type 1)	1
Hypertensive nephrosclerosis.	1
Immunoglobulin light chain deposition disease	1
Sjogren’s syndrome	1
Lupus nephritis (ISN/RPN Class III)	1

**Table 3 life-13-01492-t003:** Biopsy diagnoses in 16 patients with NDKD only at biopsy.

NDKD Pathology	*n* (Total = 16)
^a^ FSGS	4
Acute tubular necrosis	3
Hypertensive nephrosclerosis.	1
IgA nephropathy	2
Membranoproliferative/mesangiocapillary GN (type not specified)	1
Myeloma	1
Acute interstitial nephritis	1
Ischaemic nephrosclerosis	2
Anti-glomerular basement membrane disease	1

^a^ FSGS + Hypertensive nephrosclerosis in one patient and secondary FSGS in one patient.

**Table 4 life-13-01492-t004:** Clinically diagnosed and biopsy-proven diagnoses compared.

	DKD Present at Biopsy	Biopsy-Proven NDKD Only
**Clinically diagnosed DKD**	37	6
**Clinically diagnosed NDKD**	1	10

**Table 5 life-13-01492-t005:** Results of sensitivity analyses.

**Sensitivity analysis 1:** **Excluding patients with DKD as second listed clinical diagnosis (*n* = 50).**	**DKD present at biopsy**	**Biopsy proven NDKD only**
Clinically diagnosed DKD	34	5
Clinically diagnosed NDKD	1	10
**Sensitivity analysis 2:** **Excluding patients with DKD as a shared pathology at biopsy (*n* = 40).**	**DKD only present at biopsy**	**Biopsy proven NDKD only**
Clinically diagnosed DKD	24	6
Clinically diagnosed DKD	0	10

**Table 6 life-13-01492-t006:** Proteinuria levels by biopsy diagnosis.

Proteinuria Category ^a^	DKD Present at Biopsy	Biopsy-Proven NDKD Only
1	1 (2.6%)	3 (18.8%)
2	2 (5.3%)	2 (12.5%)
3	11 (28.9%)	9 (56.3%)
4	24 (63.2%)	2 (12.5%)

^a^ See Table 1 for details on proteinuria category.

**Table 7 life-13-01492-t007:** Microscopic haematuria by biopsy diagnosis.

Microscopic Haematuria Present ^a^	DKD Present at Biopsy	Biopsy-Proven NDKD Only
Yes	14 (36.8%)	3 (18.8%)
No	24 (63.2%)	13 (81.2%)

^a^ See Appendix A for details.

## Data Availability

All patient data relevant to this study have been supplied (Appendix A). The patient-level data from the parent study are unavailable due to privacy restrictions but may be made available on request by application to the CKD.QLD registry.

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
