# Peer review of "The Diagnostic Performance of a Clinical Diagnosis of Diabetic Kidney Disease"

_life, 2023, doi:10.3390/life13071492_

Round 1
Reviewer 1 Report
Methodology:
1. Authors need to provide inclusion and exclusion criteria of the study.
There is no clarity on what patients exactly were included?
Is it only patients with clinical diagnosis of DKD?
If so results show that only 79.6% are DKD clinically.
2. Details on the methodology including the processing and reporting of biopsy samples needs to be done.
How was the biopsy done and by whom? (Nephrologist/ Radiologist?). What is the technique followed? (USG guided/ Blind?).
How was the adequacy of samples analyzed?
How many pathologists are involved in the process of reporting?
3. Authors need to clarify the histopathological criteria followed for the diagnosis and classification of DKD.
4. How are Type-1 and Type-2 DM identified?
Results:
1. Why did the authors compare the study population with rest of the patients in the registry? What is the rationale behind that?
And what exactly are these 2316 patients against whom the data was compared? Did they include the patients excluded patients? If so why it was done?
There is no proper explanation in methodology or the results section on this.
2. Baseline characteristics are very incomplete. Need to provide data on the following:
i. Duration of diabetes?
ii. Weight and BMI
iii. How many are Type-1 and Type-2 DM?
iv. What is the mean proteinuria of study population?
How many patients have got nephrotic proteinuria?
What is the categorization of proteinuria followed? Reference?
v. What is the renal syndromic presentation of the study population?
AKI?/ AKI on CKD?/ RPRF/ Nephrotic syndrome?
vi. Indications for biopsy?
vii. What about other end-organ damage (DR/DSPN/CAD/PAD/CVA)?
viii. How may had microscopic hematuria?
Authors mentioned the many had microscopic hematuria but no exact numbers for the same.
ix. How many patients had HTN? What is the BP control?
x. Degree of glycemic control? Mean HbA1C?
xi. How many had extra renal manifestations suggestive of other causes of secondary GN?
xii. How many patients are on other agents retarding CKD (SGLT2i’s/ MRA’s)?
3. Most common NDKD pathology noted by the authors was secondary FSGS. Evidence for secondary FSGS? What was the evaluation done? What was the presentation of these patients and what are the causes of secondary FSGS noted?
4. 14 patients had co-existent NDKD and 38 had only DKD.
What about the other 2 patients?
5. IF the patients have co-existent NDKD on biopsy, the diagnosis should be NDKD+DKD and not DKD alone. In other words, these patients should be treated for NDKD and to be categorized under NDKD. Hence, total of 28 patients as per the author’s results have NDKD (of 54).
6. Table 2/3 and table 4 are contradictory (numbering of tables is also incorrect)
7. How was Sjogren’s syndrome diagnosed on renal biopsy? Was there anything other than tubulointerstitial inflammation seen? Why was DKD suspected if there is serological evidence of Sjogren’s? (Based on which the diagnosis was mostly made)
8. What do the authors mean by SLE (WHO grade 3) on biopsy?
Did they mean Class 3 lupus nephritis as per ISN-RPS classification?
If so, terminology needs to be changed to ISN/RPS classification terms (2018)
9. Mesangioproliferative and mesangiocapillary GN are two different pathologies.
Did the authors mean membranoproliferative GN ? If so change the terminology.
Also, the claasification of MPGN to be followed is based on IF, not as per EM.
10. Exact numbers need to be mentioned in the baseline characteristics and the percentage in the parenthesis. Only % is not acceptable.
11. Standard journal formats of expressing the Median/IQR/SD needs to be followed.
12. Standard formats to be followed for abbreviations such as CKD-EPI
13. Abbreviations in the table needs to be elaborated in the table foot note.
14. Authors need to re-analyse the data as the NDKD number needs to be re-verified.Methodology:
1. Authors need to provide inclusion and exclusion criteria of the study.
There is no clarity on what patients exactly were included?
Is it only patients with clinical diagnosis of DKD?
If so results show that only 79.6% are DKD clinically.
2. Details on the methodology including the processing and reporting of biopsy samples needs to be done.
How was the biopsy done and by whom? (Nephrologist/ Radiologist?). What is the technique followed? (USG guided/ Blind?).
How was the adequacy of samples analyzed?
How many pathologists are involved in the process of reporting?
3. Authors need to clarify the histopathological criteria followed for the diagnosis and classification of DKD.
4. How are Type-1 and Type-2 DM identified?
Results:
1. Why did the authors compare the study population with rest of the patients in the registry? What is the rationale behind that?
And what exactly are these 2316 patients against whom the data was compared? Did they include the patients excluded patients? If so why it was done?
There is no proper explanation in methodology or the results section on this.
2. Baseline characteristics are very incomplete. Need to provide data on the following:
i. Duration of diabetes?
ii. Weight and BMI
iii. How many are Type-1 and Type-2 DM?
iv. What is the mean proteinuria of study population?
How many patients have got nephrotic proteinuria?
What is the categorization of proteinuria followed? Reference?
v. What is the renal syndromic presentation of the study population?
AKI?/ AKI on CKD?/ RPRF/ Nephrotic syndrome?
vi. Indications for biopsy?
vii. What about other end-organ damage (DR/DSPN/CAD/PAD/CVA)?
viii. How may had microscopic hematuria?
Authors mentioned the many had microscopic hematuria but no exact numbers for the same.
ix. How many patients had HTN? What is the BP control?
x. Degree of glycemic control? Mean HbA1C?
xi. How many had extra renal manifestations suggestive of other causes of secondary GN?
xii. How many patients are on other agents retarding CKD (SGLT2i’s/ MRA’s)?
3. Most common NDKD pathology noted by the authors was secondary FSGS. Evidence for secondary FSGS? What was the evaluation done? What was the presentation of these patients and what are the causes of secondary FSGS noted?
4. 14 patients had co-existent NDKD and 38 had only DKD.
What about the other 2 patients?
5. IF the patients have co-existent NDKD on biopsy, the diagnosis should be NDKD+DKD and not DKD alone. In other words, these patients should be treated for NDKD and to be categorized under NDKD. Hence, total of 28 patients as per the author’s results have NDKD (of 54).
6. Table 2/3 and table 4 are contradictory (numbering of tables is also incorrect)
7. How was Sjogren’s syndrome diagnosed on renal biopsy? Was there anything other than tubulointerstitial inflammation seen? Why was DKD suspected if there is serological evidence of Sjogren’s? (Based on which the diagnosis was mostly made)
8. What do the authors mean by SLE (WHO grade 3) on biopsy?
Did they mean Class 3 lupus nephritis as per ISN-RPS classification?
If so, terminology needs to be changed to ISN/RPS classification terms (2018)
9. Mesangioproliferative and mesangiocapillary GN are two different pathologies.
Did the authors mean membranoproliferative GN ? If so change the terminology.
Also, the claasification of MPGN to be followed is based on IF, not as per EM.
10. Exact numbers need to be mentioned in the baseline characteristics and the percentage in the parenthesis. Only % is not acceptable.
11. Standard journal formats of expressing the Median/IQR/SD needs to be followed.
12. Standard formats to be followed for abbreviations such as CKD-EPI
13. Abbreviations in the table needs to be elaborated in the table foot note.
14. Authors need to re-analyse the data as the NDKD number needs to be re-verified.
English is ok and however, needs to be consistent with standard terminology used across the manuscript.
Author Response
Dear reviewer 1,
Thank you for your detailed review and for giving us the opportunity to respond rather than rejecting the manuscript.
I have addressed your comments in order in the attached word document.

Reviewer 2 Report
Since DKD is not an indication for immunosuppressive therapy, kidney biopsy is not required. Therefore, this is a very unique study in which the sensitivity and specificity of the clinical diagnosis of DKD is compared to the results of subsequent kidney biopsies.
Minor;
The paper discusses diabetes duration, diabetic retinopathy, and microscopic hematuria as predictive markers of DKD, but I would like to see more discussion of other factors as well. For example, HbA1C, glycoalbumin, and diabetic neuropathy should also be discussed.
Minor editing of English language required.
Author Response
Dear reviewer 2,
Thank you very much for your review. I have responded to your comments in the attached document and also hope that these will be visible in the updated version of the manuscript.
